# Prehospital delay is an important risk factor for mortality in community-acquired bloodstream infection (CA-BSI): a matched case–control study

Martin Holmbom  ,[1,2] Maria Andersson,[2] Sören Berg,[3] Dan Eklund,[2] Pernilla Sobczynski,[2] Daniel Wilhelms,[4] Anna Moberg,[5] Mats Fredrikson,[6] Åse Östholm Balkhed,[2] Håkan Hanberger[2]

ÅÖB and HH contributed equally.

For numbered affiliations see end of article.

**Correspondence to**
Dr Martin Holmbom;
martin.holmbom@
regionostergotland.se

## ABSTRACT

**Objectives** The aim of this study was to identify prehospital and early hospital risk factors associated with 30-day mortality in patients with blood culture-confirmed community-acquired bloodstream infection (CA-BSI) in Sweden.

**Methods** A retrospective case–control study of 1624 patients with CA-BSI (2015–2016), 195 non-survivors satisfying the inclusion criteria were matched 1:1 with 195 survivors for age, gender and microorganism. All forms of contact with a healthcare provider for symptoms of infection within 7 days prior CA-BSI episode were registered. Logistic regression was used to analyse risk factors for 30-day all-cause mortality.

**Results** Of the 390 patients, 61% (115 non-survivors and 121 survivors) sought prehospital contact. The median time from first prehospital contact till hospital admission was 13 hours (6–52) for non-survivors and 7 hours (3–24) for survivors (p<0.01). Several risk factors for 30-day all-cause mortality were identified: prehospital delay OR=1.26 (95% CI: 1.07 to 1.47), p<0.01; severity of illness (Sequential Organ Failure Assessment score) OR=1.60 (95% CI: 1.40 to 1.83), p<0.01; comorbidity score (updated Charlson Index) OR=1.13 (95% CI: 1.05 to 1.22), p<0.01 and inadequate empirical antimicrobial therapy OR=3.92 (95% CI: 1.64 to 9.33), p<0.01. In a multivariable model, prehospital delay >24 hours from first contact remained an important risk factor for 30-day all-cause mortality due to CA-BSI OR=6.17 (95% CI: 2.19 to 17.38), p<0.01.

**Conclusion** Prehospital delay and inappropriate empirical antibiotic therapy were found to be important risk factors for 30-day all-cause mortality associated with CA-BSI. Increased awareness and earlier detection of BSI in prehospital and early hospital care is critical for rapid initiation of adequate management and antibiotic treatment.

## BACKGROUND

Bloodstream infection (BSI) is a major cause of morbidity and mortality worldwide, with sepsis and septic shock as important complications.[1–5] In Europe, the number of BSIs and number of deaths due to BSI each year have been estimated to be 1.2 million and 157 000, respectively.[6] Most studies in Nordic countries report an increase in the incidence of BSI[7–10] and an increase in associated mortality,[7 9] but a low level of antibiotic resistance.[11]

The increasing incidence of community-onset BSI has become a major problem. This may be due to an increase in healthcare-associated BSI related to complex medical care of aged patients with comorbidity in the community setting.[7 12] BSI and the risk of transition to sepsis and septic shock with life-threatening organ dysfunction requires early identification for prompt initiation of life-saving measures and improvement in outcome.[13–17] There are many studies focusing on improving early detection of patients at risk for developing sepsis in the emergency department (ED).[18–22] Few studies have addressed prehospital care of sepsis prior to evaluation at the ED or by the paramedic,[23 24] and not one, to our knowledge, on the impact of delay in prehospital care on 30-day mortality in a well-defined case–control study on patients with community-acquired BSI (CA-BSI).

**STRENGTHS AND LIMITATIONS OF THIS STUDY**

⇒ Non-surviving patients were matched 1:1 with survivors for age, gender and microorganism to evaluate risk factors for 30-day mortality.
⇒ Data on limitation of level of care were collected and evaluated as a potential confounding factor.
⇒ Only empirical antibiotic treatments were analysed, early changes in therapy were not considered.
⇒ Data regarding patient delay, that is, time from onset of symptoms to contact with prehospital or hospital care, were unavailable.

The aim of this study was to identify prehospital and early hospital risk factors associated with 30-day mortality in patients with culture-confirmed CA-BSI.

## METHODS
### Study design and setting
This was a case–control study on risk factors for 30-day mortality in patients with CA-BSI. From a database of culture-confirmed CA-BSI in the Swedish county of Östergötland, samples of cases (non-survivors) and matched controls (survivors) over a 2-year period (from 1 January 2015 until 31 December 2016) were selected. The county of Östergötland, with a population of 445 000 in December 2015 and 452 000 in December 2016, had four hospitals, one tertiary care university hospital, two general hospitals, one district hospital, 44 primary healthcare centres (PHCCs) and four out-of-office hours centres.

In most West European countries, prehospital healthcare is available at several levels. In this study, prehospital care is defined as healthcare contact prior to transport by ambulance or entering the ED. In Sweden, first contact with the healthcare service is usually by a phone call to the PHCC or to the national healthcare guide (NHG). In urgent situations, the emergency medical service (i.e, paramedic ambulance service) is contacted directly. PHCCs usually provide a call-back system where a nurse rings during office hours and assesses the need for clinical assessment. The NHG offers immediate medical advice over the phone 24 hours a day. It is manned by specially trained nurses using a decision-support algorithm, they assess the need for further care and provide advice and/or recommend other healthcare services.

### Data collection
From the microbiology laboratory database, the following dataset was collected: blood culture results, number of aerobic and anaerobic blood culture vials taken, sites of puncture, species identification and susceptibility patterns. The dataset was entered into a second database where it was linked to the patient-administration system providing the following data on all patients with a BSI episode: gender, age, comorbidity, admitting department, date of admission, date of discharge and mortality.

A total of 2356 BSI episodes were identified, of which 1624 were CA-BSIs. To determine the impact of prehospital and early hospital care on 30-day all-cause mortality 2015–2016, we matched a group of non-survivors (cases) with survivors (controls) that satisfied the following inclusion criteria: adult (≥18 years), culture-confirmed CA-BSI with a significant pathogen, registered in the County of Östergötland, treated in a hospital in the County of Östergötland, non-survivor (dead within 30 days from positive blood culture) and survivor alive at (30 days).

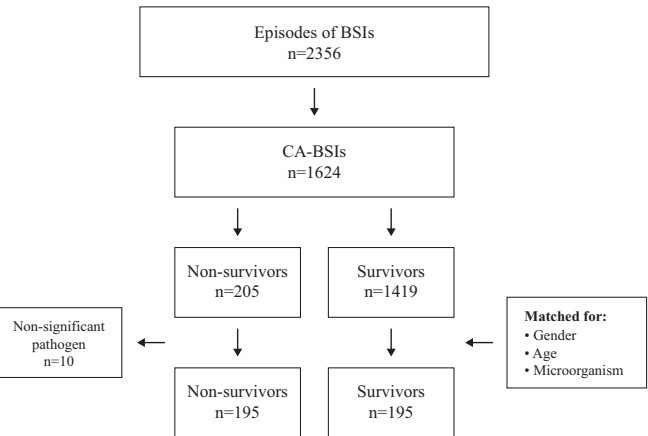

**Figure 1** A total of 2356 BSI episodes were identified, of which 1624 were CA-BSIs, 195 non-survivors met the inclusion criteria and were matched with 195 survivors for age (±10 years), gender and microorganisms. CA-BSI, community-acquired bloodstream infection.

In all, 195 non-survivors met the inclusion criteria and were matched with 195 survivors for age (±10 years), gender and microorganisms (figure 1).

For non-survivors and survivors, data on all infection-related contacts with a healthcare provider (telephone contact or physical contact) ≤7 days prior to a positive blood culture were collected by reviewing medical records. For each contact, the following were documented in the case report form: profession of healthcare provider, suspected diagnosis, vital signs, primary focus of infection and initial advice given. In the event of ambulance transport, information on time to reach the patient, time to reach ED, vital signs, intravenous fluid administration and oxygen administration were extracted from the mission report.

The following data on management in the ED were collected: limitation of level of care, medical personnel involved, vital signs, laboratory data, consultation with a specialist if any (infectious diseases, internal medicine, anaesthesia and so on), time on the ED, times to fluid administration and antibiotic treatment, other sepsis treatment such as corticosteroids and highest serum lactate within 24 hours after admission. The highest priority given by the triage system Rapid Emergency Triage and Treatment System (RETTS) was documented.[25 26] Sequential Organ Failure Assessment (SOFA) score[27] and National Early Warning Score (NEWS) 2[28] were also calculated.

An expert group consisting of physicians specialised in infectious diseases judged the correctness of the antibiotic treatment based on local recommendations for empirical treatment of the suspected focus at that time, and the pathogen finally cultured. Doses of antibiotics, corrected for renal function, were also judged. Comorbidity scores according to the modified Charlson Comorbidity Index, based on diagnoses obtained from patient medical records up to 24 months prior to the BSI, were later entered into the database.

## Definitions

### Blood culture
One set of blood cultures comprised one aerobic and one anaerobic blood culture bottle. It was recommended that at least two sets of blood cultures be taken simultaneously.

### Positive blood culture
The isolations of microorganism(s) (one or more bacterial or fungal isolates) from a set of blood cultures. Only bacterial or yeast isolates from initial blood cultures were considered; repeat isolates were excluded.

### Non-significant pathogens
Microorganisms typically belonging to the skin microbiome

(Coagulase-Negative Staphylococci (CoNS), *Micrococcus* spp, *Bacillus* spp, *Corynebacterium* spp and *Propionibacterium* spp) were considered probable contaminants and excluded.[29] An exception was CoNS isolated from blood cultures taken from at least two different puncture sites on the same occasion, and with an identical resistance pattern.

### Community-acquired bloodstream infection
A positive blood culture obtained within 48 hours after hospital admission. Cultures from readmission within 30 days were excluded. Furthermore, patients who had received intravenous treatment or advanced nursing care at home such as parenteral nutrition, haemodialysis or intravenous chemotherapy, within the previous 30 days were excluded.

### Immunosupression
Patients were deemed immunocompromised if prescribed immunosuppressive medication or had a code number (D80–D89) for an immunosuppressive disease according to the International Classification of Diseases, 10th edition (ICD-10) when admitted to the hospital.

### Comorbidity
Comorbidity score was based on the updated Charlson Comorbidity Index[30–32] using ICD-10 diagnosis codes documented in the patient's medical record up to 24 months prior to date of admission (online supplemental table 1).

### Sepsis
The Sepsis-3 definition[33] was used, that is, an increase by two points or more in the SOFA score due to infection.

### Limitation of level of care
In some cases, the decision was made to limit the level of care. Such a decision influences treatment provided and thus mortality. This is an obvious confounding factor. Limitation of level of care set in the ED or within the first 24 hours of admission was registered as: (a) withholding of intensive care; (b) withholding of cardiopulmonary resuscitation (CPR) and (c) withdrawal of all active treatment, that is, end-of-life care (EoLC).

### Appropriate empirical antibiotic therapy
Defined by local recommendations, suspected source of infection, severity of illness, dosage and correction for renal function.[34 35]

### Microbiologically appropriate empirical antibiotic therapy
Defined according to cultured pathogen and its susceptibility patterns.[34–36]

## Microbiological methods
All microorganisms isolated were analysed at the species level. Microorganism identification and susceptibility testing were performed at the Clinical Microbiology Department, University Hospital, Linköping using matrix-assisted laser desorption ionisation time-of-flight mass spectrometry. Antibiotic susceptibility classification used during study period was: susceptible (S), intermediate (I) and resistant (R), according to the European Committee on Antimicrobial Susceptibility Testing guidelines.[36]

## Statistical analysis
Data are presented as mean and SD or median and quartiles (Q1–Q3). We compared normally distributed quantitative variables using the Student's t-test and categorical variables using the $\chi^2$ test or Fisher exact test as appropriate. Mann-Whitney U test and Kruskal-Wallis test were used otherwise. A two-tailed p value <0.05 was considered statistically significant. Correlations were analysed according to Pearson. Missing data were treated by pairwise deletion, and there was no imputation. Univariate and multivariable analyses were performed with logistic regression including the descriptive parameters that correlated to 30-day mortality. Two patients among the non-survivors had EoLC taken prior to admission to the hospital. This was considered to have had a major effect on early treatment and time to admission to hospital for these patients and they and their matched survivors were excluded from the analyses regarding time to admission and the multiple regression model. The statistics programme SPSS V.25 was used for the analyses.

## RESULTS

### Demographic and clinical characteristics
A total of 2356 BSI episodes were identified during the study period, of which 1624 (69%) were CA-BSIs. The all-cause 30-day mortality rate of CA-BSI was 12.6% (n=205). Of these, 95% (n=195) met the inclusion criteria and were included in the study as non-survivors (case-group) and were matched with 195 survivors (control group) for gender, age and microorganism. Almost 40% of the non-survivors died within 3 days after admission, and 80% within 7 days. Underlying comorbidity scores according to the updated Charlson Comorbidity Index were higher among non-survivors than survivors (3.2 vs 2.3, p=<0.01). Non-survivors were more often diagnosed with cancer (34% vs 23%, p=0.01) and metastatic carcinoma (14% vs 7%, p=0.02) compared with survivors. Limitation of level

**Table 1** Demographics and prehospital data

| | Non-survivors, n=195 | Survivors, n=195 | P value |
|---|---|---|---|
| Demographics (%) | | | |
| Male | 106 (54) | 106 (54) | >0.99 |
| Mean age (SD) | 78 years (±13) | 76 years (±12) | 0.20 |
| Surgery within 30 days | 29 (15) | 27 (14) | 0.77 |
| Immunosuppression | 60 (31) | 45 (23) | 0.09 |
| Charlson (update weight) (SD) | 3.2 (2.99) | 2.3 (2.36) | *0.01* |
| Patients with any limitation of level of care before admission (%) | 28 (14) | 4 (2) | *<0.01* |
| EoLC | 2 (1) | 0 | 0.50 |
| No ICU | 17 (9) | 2 (1) | *<0.01* |
| No CPR | 27 (14) | 4 (2) | *<0.01* |
| First prehospital contact site (%) | | | |
| Phone to NHG | 14 (7) | 60 (31) | *<0.01* |
| Phone to a PHCC | 79 (41) | 33 (17) | *<0.01* |
| Patient visit to a PHCC | 22 (11) | 28 (14) | 0.36 |
| No prehospital contact | 80 (41) | 74 (38) | 0.53 |
| Multiple prehospital contacts (≥2) (%) | 64 (33) | 41 (21) | *0.01* |
| Reason for first prehospital contact (number) (%) | n115 | n121 | |
| Fever | 44 (38) | 43 (36) | 0.67 |
| Chills | 13 (11) | 13 (11) | 0.89 |
| 'Found on the floor' | 4 (3) | 0 | *0.04* |
| Gastrointestinal symptoms | 16 (14) | 23 (19) | 0.29 |
| Cough | 3 (3) | 13 (11) | *0.01* |
| Dyspnoea/breathing difficulties | 23 (20) | 15 (12) | 0.11 |
| Urinary tract symptoms | 14 (12) | 22 (18) | 0.20 |
| Rapid deterioration in general condition | 47 (41) | 29 (24) | *<0.01* |
| Fatigue | 3 (3) | 15 (12) | *<0.01* |
| Time from first prehospital contact to admission, hours, median (Q1–Q3) | 13 (5.9–51.6) n113 | 7.2 (3.3–24) n119 | *<0.01* |
| Phone to NHG | 4 (1.0–8.7) n13 | 3.6 (1.3–7.3) n58 | 0.64 |
| Phone to a PHCC | 24 (6.8–72) n78 | 12.5 (5.9–24) n33 | 0.06 |
| Visit to a PHCC | 11 (6.6–77) n22 | 12.5 (6.4–66) n28 | 0.75 |
| Time from first prehospital contact to admission, intervals (%) | n193 | n193 | |
| 0–6 hours | 29 (15) | 49 (25) | *0.01* |
| 6–12 hours | 26 (14) | 30 (16) | 0.56 |
| 12–24 hours | 12 (6) | 21 (11) | 0.10 |
| >24 hours | 46 (24) | 19 (10) | *<0.01* |
| No prehospital contact | 80 (41) | 74 (38) | 0.53 |

Data are presented as no. (%) or mean (SD).
Pearson $\chi^2$, Fisher's exact test or t-test, as appropriate. P values <0.05 are shown in italics. Time indications are calculated with median, interquartile 25th to 75th percentile range (Q1–Q3) and Mann-Whitney U. Reason for first prehospital contact: more than one reason possible.
CPR, cardiopulmonary resuscitation; EoLC, end-of-life care; ICU, intensive care unit; NHG, national healthcare guide; PHCC, primary healthcare centre.

of care before admission was significantly more common among non-survivors (14% vs 2%, p=<0.01), (table 1 and online supplemental table 2).

**Microbiology**

There were 407 isolates from 390 cases. *Staphylococcus aureus* was the most frequently isolated primary pathogen

(27%), followed by *Escherichia coli* (21%), *Streptococcus pneumoniae* (8%) and *Klebsiella pneumoniae* (7%). *Candida* spp were observed in 2% of cases. Polymicrobial CA-BSI was found in nine non-survivors and eight survivors. Microbiological findings are summarised in online supplemental table 3.

## Prehospital data

Of the 390 patients, 61% (115 non-survivors and 121 survivors) had taken contact with a prehospital healthcare facility for infection-related symptoms within 7 days prior to admission to hospital. Forty-four per cent of these patients had ≥2 prehospital contacts. Forms of prehospital contact were phone contact with a PHCC (47%), phone contact with NHG (31%) and visit to a PHCC (21%). Fever was the primary reason for prehospital contact (37%). Rapid deterioration in general condition was the second most common reason (32%) and was more common among non-survivors than survivors (41% vs 24%, p<0.02) (table 1). Phone contact with a PHCC was the more common form of first contact among non-survivors (41% non-survivors vs 17% survivors, p=0.01), whereas phone contact with NHG was more common among survivors (7% non-survivors vs 31% survivors, p=<0.01) (table 1).

## Time from first infection-related prehospital contact till hospital admission

In total, the median time from first prehospital contact till hospital admission was 8 hours (4–27). This was significantly longer for non-survivors 13 hours (6–52) versus 7.2 hours (3–24) for survivors, p=0.01 (table 1). For the study group as a whole, there were significant differences in time to admission depending on form of first contact, with shorter median time to admission after telephone contact with the NHG 3.8 hours (1.3–7.7) compared with phone contact with a PHCC 24.0 hours (6.8–48) or visit to a PHCC 11.3 hours (6.7–72) (p<0.01).

## Hospital data

A total of 88% were admitted to hospital via the ED (85% non-survivors vs 91% survivors, p=0.04), other patients were admitted via a specialist outpatient department. Hospital admission after ambulance transport was more common among non-survivors (81% vs 70%, p=0.01) (table 2). The most common sign reported when admitted to hospital was fever (54%). The most common infection focus was a urinary tract infection (24%) followed by infection with unknown focus (23%) and respiratory tract infection (22%). There were no differences between non-survivors and survivors regarding primary focus of infection (online supplemental table 4).

On the ED, vital signs and most laboratory test results indicated a more serious condition in non-survivors compared with survivors and 159 non-survivors and 161 survivors were triaged with a life-threatening or potentially life-threatening condition according to the RETTS score. Abnormalities in respiratory function,

haemodynamics and neurological function were significantly more prominent among non-survivors (online supplemental table 4). SOFA score on admission and at 24 hours was higher among non-survivors 4.2 (SD: 2.3) versus 2.3 (SD: 1.7) (p<0.01) and 6.8 (3.6) versus 3.8 (2.7) (p<0.01), respectively. On admission to hospital, 75% of non-survivors and 52% of survivors (p<0.01) fulfilled the Sepsis 3 criteria, and within 24 hours the corresponding figures were 95% of non-survivors and 79% of survivors (p<0.01). In all, 11% of non-survivors received intensive care unit (ICU) care within 24 hours compared with 5% of survivors (p=0.01) (table 2).

Antibiotic treatment was initiated in 96% of non-survivors and 99.5% of survivors, (p=0.07). The median time from hospital admission to start of empirical antibiotic did not differ between the groups, non-survivors 2.8 hours (1.4–5.3) and survivors 3.0 hours (1.4–6.2), nor in the case of sub-classification according to triage score (RETTS) (online supplemental table 5). Antibiotic administration was started within 1 hour in 18% of non-survivors and 16% of survivors (p=0.48). More survivors received appropriate empirical antibiotic therapy than non-survivors (87% non-survivors and 96% survivors, p<0.01). Microbiologically appropriate empirical antibiotic therapy did not differ significantly between the groups (82% non-survivors vs 88% survivors) (table 2). The empirical antibiotics used are listed in online supplemental table 6.

## Analysis of risk factors for 30-day mortality in CA-BSI

In the univariate logistic regression, several risk factors associated with 30-day mortality were observed: comorbidity score (updated Charlson Index) OR=1.13 (95% CI 1.05 to 1.22), p<0.01, prehospital delay OR=1.26 (95% CI 1.07 to 1.47), p=0.01, severity of illness (SOFA score on admission) OR=1.60 (95% CI 1.40 to 1.83), p<0.01 and inappropriate empirical antibiotic therapy OR=3.92 (95% CI 1.64 to 9.33), p<0.01 (table 3).

In the multivariable model of 30-day mortality, prehospital delay >24 hours was significantly related to mortality, OR=6.17 (95% CI 2.19 to 17.38), p<0.01. This was followed by inappropriate empirical antibiotic therapy, OR=5.50 (95% CI 1.62 to 18.63), p=0.01, no prehospital contact, OR=2.56 (95% CI 1.02 to 6.41), p=0.045 and severity of illness (SOFA score on admission), OR=1.35 (95% CI 1.16 to 1.57), p<0.01 (table 3).

## DISCUSSION

This study revealed that prehospital delay had a major impact on mortality among patients with CA-BSI. Non-survivors were more seriously ill when admitted to hospital, with significantly higher SOFA scores and more signs of sepsis. Although initial care at the hospital and time to antibiotic treatment were comparable between non-survivors and survivors (for some variables even better among non-survivors) this was not enough to compensate for the negative effect of prehospital delay.

**Table 2** Hospital data—severity of diseases and antibiotic treatment

| | Non-survivors, n=195 | Survivors, n=195 | P value |
|---|---|---|---|
| Ambulance transport (%) | 158 (81) n194 | 136 (70) n193 | *0.01* |
| Admission to hospital (%) | | | |
| Emergency department | 165 (85) | 178 (91) | *0.04* |
| Specialist outpatient department | 30 (15) | 17 (9) | *0.04* |
| Severity of disease (SD) | | | |
| Habitual SOFA | 0.8 (1.1) n185 | 0.4 (0.8) n193 | *<0.01* |
| SOFA score on admission | 4.2 (2.3) n161 | 2.3 (1.7) n160 | *<0.01* |
| SOFA score at 24 hours | 6.8 (3.6) n187 | 3.8 (2.7) n195 | *<0.01* |
| Maximum lactate level mmol/l first 24 hours | 4.6 (4.0) n122 | 2.9 (1.9) n110 | *<0.01* |
| NEWS 2 | 7.1 (4.0) n163 | 5.0 (3.4) n181 | *<0.01* |
| Sepsis (%) | | | |
| Sepsis on admission | 119 (75) n158 | 82 (52) n158 | *<0.01* |
| Sepsis at 24 hours | 176 (95) n185 | 152 (79) n193 | *<0.01* |
| ICU care within 24 hours (%) | 22 (11.3) | 9 (4.6) | *0.01* |
| Antibiotic treatment | | | |
| Antibiotic administration (%) | 186 (96) n193 | 192 (99.5) n193 | 0.07 |
| Time (hours) to antibiotics from admission median, (Q1–Q3) | 2.8 (1.4–5.3) n185 | 3.0 (1.4–6.2) n192 | 0.58 |
| First dose antibiotic within 1 hour (%) | 34 (18) n185 | 30 (16) n192 | 0.48 |
| Appropriate empirical antibiotic therapy (%) | 162 (87) n186 | 185 (96) n192 | *<0.01* |
| Microbiologically appropriate empirical antibiotic therapy (%) | 155 (83) n186 | 168 (88) n192 | 0.25 |
| Intravenous fluids in the ED (%) | 124 (76) n163 | 116 (68) n171 | 0.09 |

Data are presented as no. (%) or mean (SD).

Pearson $\chi^2$, Fisher's exact test or t-test, as appropriate. P values <0.05 are shown in italics. Time indications are calculated with median, interquartile 25th to 75th percentile range (Q1–Q3) and Mann-Whitney. Two non-survivors with end-of-life care (EoLC) on arrival at hospital, and their controls were excluded from the analysis regarding antibiotic treatment and intravenous fluids.

ED, emergency department; ICU, intensive care unit; NEWS 2, National Early Warning Score 2; SOFA, Sequential Organ Failure Assessment.

Patients with CA-BSI form a heterogeneous group in which underlying disease, gender, age and appropriate management vary significantly. The severity of BSI ranges from asymptomatic bacteraemia to fulminant sepsis or septic shock. To our knowledge, this is the first study to focus on a well-defined population of adult patients with CA-BSI focusing on prehospital care prior to hospital admission. In previous study by Holmbom *et al*, gender, age and pathogen were risk factors for mortality in BSI,[7 11] so these were used when matching patients to enable evaluation of other potential risk factors in this study. Compared with previous studies and reports on sepsis and CA-BSI, the distribution of pathogens in the present study was different,[7 37 38] with *S. aureus* being the most common. This was probably a result of the study design, that is, inclusion of patients dying within 30 days and matched controls, since *S. aureus* BSI has such a high mortality rate.[39 40]

Delay in antibiotic treatment has previously been shown to have a clear association with increased mortality in progressive sepsis and septic shock.[13 15 17 37 41 42]

However, delay prior to ambulance transport or coming to the ED, and causes of delay have hardly been investigated.[24] Strategies for early identification of sepsis in prehospital care and measures to increase survival from sepsis have received much attention, and several strategies and screening tools have been investigated with varied results.[18 21 43] Among the patients in the present study who took prehospital contact with the healthcare service, form of first prehospital contact affected prehospital delay. Telephone contact with the NHG was associated with shorter time to hospital admission than phone contact or visit to the local PHCC. The reason for this remains unclear. Possible factors include: (1) compliance to the NHG computer-based decision algorithm designed for acute illness, as well as nurses more experienced in giving advice over the phone; (2) more liberal attitude of NHG nurses to send patients to the ED; (3) non-survivors have higher comorbidity scores and tend to seek initial contact with their PHCC. Frail patients have also greater difficulty in describing their symptoms, making it more difficult for the primary healthcare provider to evaluate

**Table 3** Risk factors for 30-day mortality

| Risk factor | Univariate analysis | | | Multivariable analysis | | |
|---|---|---|---|---|---|---|
| | OR | 95% CI | P value | OR* | 95% CI | P value |
| Time from first prehospital contact to admission (intervals) | 1.26 | 1.07 to 1.47 | *0.01* | | | |
| 0–6 hours | 1† | | | 1† | | |
| 6–12 hours | 1.46 | 0.73 to 2.94 | 0.28 | 1.35 | 0.48 to 3.79 | 0.56 |
| 12–24 hours | 0.97 | 0.42 to 2.25 | 0.94 | 0.56 | 0.15 to 2.10 | 0.39 |
| >24 hours | 4.09 | 2.02 to 8.28 | *<0.01* | 6.17 | 2.19 to 17.38 | *<0.01* |
| No prehospital contact | 1.83 | 1.05 to 3.19 | *0.03* | 2.56 | 1.02 to 6.41 | 0.05 |
| Updated Charlson | 1.13 | 1.05 to 1.22 | *<0.01* | 1.02 | 0.91 to 1.15 | 0.70 |
| Ambulance transport | 1.84 | 1.14 to 2.97 | *0.01* | 1.59 | 0.68 to 3.71 | 0.29 |
| Admission through ED | 0.55 | 0.29 to 1.03 | 0.06 | 1.35 | 0.43 to 4.25 | 0.61 |
| ln SOFA score | 1.60 | 1.40 to 1.83 | *<0.01* | 1.35 | 1.16 to 1.57 | *<0.01* |
| Inappropriate empirical antibiotic therapy‡ | 3.92 | 1.64 to 9.33 | *<0.01* | 5.50 | 1.62 to 18.63 | *0.01* |
| Rapid deterioration in general condition | 1.79 | 1.06 to 3.02 | *0.03* | 2.27 | 0.98 to 5.28 | 0.06 |
| Any care restrictions before or within 24 hours after admission | 9.95 | 5.95 to 16.64 | *<0.01* | 10.37 | 4.96 to 21.67 | *<0.01* |
| Matching | 1 | 0.99 to 1.00 | >0.99 | 1 | 0.99 to 1.0 | 0.20 |

P values <0.05 are shown in italics.
*Multivariable binomial regression analysis.
†Reference.
‡Clinical inappropriateness defined by local recommendations, suspected source of infection, severity of illness, dosage and correction for renal function.
ED, emergency department; SOFA, Sequential Organ Failure Assessment.

the seriousness of their condition and (4) whereas the NHG gives advice directly, it is possible for the PHCC to delay clinical assessment until later the same or next day. In fact, the call-you-back system used by PHCCs may have caused even more delay than seen in this study since we did not have access to when the first phone call from the patient was made. Instead, we used the time the phone-call back to the patient was made. In conclusion, the BSI healthcare process and its outcome varies according to the form of healthcare provider the patient first contacts.

We also looked for factors facilitating earlier identification of patients with increased risk for death. The comorbidity score (updated Charlson Comorbidity Index) was higher among non-survivors, with higher rates of cancer and metastatic carcinoma. These conditions are associated not only with increased mortality risk in general but also with BSI itself.[7 11 44] In prehospital records, 'found on the floor' and 'rapid deterioration in general condition' were more often reported among non-survivors. Since both reflect a seriously ill person, the patient is by proxy identified as a patient at risk. Although early prehospital identification of patients at high risk for sepsis or mortality is complex,[43] we confirm the findings of previous studies[7 11 44] that taking signs and symptoms of infection in cancer patients seriously is crucial. Otherwise, we found no patient-specific factors that could be of help in identifying patients with increased mortality risk.

Many studies on sepsis, especially prior to Sepsis 3, have had patients with a wide spectrum of disease severity. This has certainly contributed to differences in results regarding the impact of time to antibiotic treatment on outcome.[13–15 17] Our study comprised patients with CA-BSI, not true sepsis, and thus even more likely to be heterogeneous. Even so, 75% of non-survivors and 52% of survivors had sepsis on admission to the ED. The severity of illness on admission to hospital was greater in non-survivors and there are several factors that might explain this. One factor could be a longer prehospital delay as observed in this study.

Previous studies have shown the importance of rapid detection and initiation of adequate antibiotic treatment in sepsis and especially septic shock.[13–15 17 37] Nevertheless, using the RETTS triage system,[25 33] only half of the non-survivors were captured by the highest priority (red). Furthermore, priorities red plus orange together would still have failed to detect one third. Likewise, when using NEWS 2 with the proposed cut-off score of 5,[28 45] at least one third of the non-survivors would have remained undetected as patients at risk for sepsis in the initial triage. Triage is useful when escalating the care of patients already identified with sepsis, as shown by Rosenqvist et al,[22] but we lack a valid, simple, highly predictive scoring system to detect sepsis.[46 47]

Surviving Sepsis Guidelines 2016 recommend that antibiotics be administered within 1 hour of diagnosing sepsis.[48] This was not achieved in our material where only 16% of non-survivors and 18% of survivors received the first dose of antibiotics within 1 hour. Appropriate empirical antibiotic therapy according to local guidelines was more common among survivors (p<0.01), while microbiologically appropriate empirical antibiotic therapy was not. This could indicate that not only correct empirical antibiotics is important, but that also a holistic view of the patient, including suspected focus of infection and need of source control are important parts of early management. This study showed that, prehospital delay and non-appropriate empirical treatment among patients with CA-BSI worsens the outcome and measures should be taken both prehospital and at the emergency unit to increase survival in CA-BSI.

## Limitations

To our knowledge, this is the first study on a well-defined population of adult patients with CA-BSI, focusing on prehospital care prior to hospital admission, but it has limitations.

1. Though non-survivors and survivors were matched for gender, age (±10 years) and microorganism, microorganism resistance patterns were not considered when matching non-survivors and survivors. However, since the prevalence of resistant bacteria in Sweden is low, this could only have had a minor influence on the results. Furthermore, when analysing correct antimicrobial therapy based on microorganisms cultured and their resistance patterns, there was no significant difference between the groups.
2. We only analysed the correctness of empirical antibiotic treatment, while early changes in therapy were not considered.
3. Other causes of death such as myocardial infarction, respiratory failure or pulmonary embolism as primary cause of death were possible, but such conditions would likely be related to the underlying infection, and since 95% of the cases had a fulminant sepsis within 24 hours this would be the major predisposing factor.
4. The only differences in prehospital management affecting outcome in this study were the timing of events. Other variables require further research.
5. The only time parameters for early hospital care in this study were time to empirical antibiotic treatment and time for ambulance transport. Presumably, time to antibiotic treatment also implies time to early hospital treatment. Furthermore, time to early hospital care is usually based on severity of illness (RETTS and NEWS 2) when admitted to hospital. Time to antibiotic treatment based on RETTS did not differ between the groups. This study was not designed to validate the time to care based on triage level, RETTS or NEWS 2 correctness.
6. Community healthcare in Sweden has several healthcare providers, that is, care provided by the community, private medicine and/or self-medication (by patient or relative). This results in unavailable data, and it is possible that some community-onset healthcare-associated BSIs may have occurred in the study population, even though hospital and healthcare-related infections were excluded as far as possible.
7. Data regarding patient delay, that is, time from onset of symptoms to contact with prehospital or hospital care should be included when calculating hospital and therapy delay. However, these were unavailable since patient delay was not systematically registered in the patient records.

## CONCLUSION

In this case–control study, prehospital delay and inappropriate empirical antibiotic therapy were found to be important risk factors for 30-day all-cause mortality associated with CA-BSI. Effective guidelines to aid recognition of patients likely to develop sepsis in prehospital and early hospital care, and an effective prehospital medical advice service, could help to make significant progress in the care of patients with BSI.

**Author affiliations**
[1]Department of Urology, and Department of Biomedical and Clinical Sciences, Linköping University, Linköping, Sweden
[2]Department of Infectious Diseases, and Department of Biomedical and Clinical Sciences, Linköping University, Linköping, Sweden
[3]Division of Cardiothoracic Anesthesia and Intensive Care, Department of Medicine and Health Science, Faculty of Medicine and Health Sciences, Linköping University, Linköping, Sweden
[4]Department of Emergency Medicine in Linköping, and Department of Biomedical and Clinical Sciences, Linköping University, Linköping, Sweden
[5]Department of Health, Medicine and Caring Sciences, Linköping University, Linkoping, Sweden
[6]Department of Biomedical and Clinical Sciences and Forum Östergötland, Faculty of Medicine and Health Sciences, Linköping University, Linköping, Sweden

**Acknowledgements** Thanks to consultant anaesthetist Peter Cox for proof reading.

**Contributors** MH, HH and ÅÖB conceived and designed the study. MH, MA, DE, ÅÖB and MF analysed the data. MH and MA wrote the manuscript. All authors (MH, MA, SB, DE, PS, DW, AM, MF, ÅÖB and HH) contributed to the discussion and reviewed the manuscript. All authors (MH, MA, SB, DE, PS, DW, AM, MF, ÅÖB and HH) commented and approved the final version of the paper. All authors (MH, MA, SB, DE, PS, DW, AM, MF, ÅÖB and HH) contributed to manuscript. MH, MA, ÅÖB and HH is guarantor for the study.

**Funding** This work was supported by Östergötland Count Council (Award/grant number: N/A).

**Competing interests** None declared.

**Patient consent for publication** Not applicable.

**Ethics approval** The Regional Ethics Review Board in Linköping, Sweden, approved the study (Ref. no: 2010/160-31).

**Provenance and peer review** Not commissioned; externally peer reviewed.

**Data availability statement** All data relevant to the study are included in the article or uploaded as supplemental information. All relevant data are within the manuscript and its Supplementary files.

**ORCID iD**
Martin Holmbom http://orcid.org/0000-0002-3706-2294

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
