## [Reviewer comments · BMJ Open]

ARTICLE DETAILS

TITLE (PROVISIONAL)	Prehospital delay is an important risk factor for mortality in community-acquired bloodstream infection (CA-BSI): A matched case-control study
AUTHORS	Holmbom, Martin; Andersson, Maria; Berg, Sören; Eklund, Dan; Sobczynski, Pernilla; Wilhelms, Daniel; Moberg, Anna; Fredrikson, Mats; Balkhed, Åse Östholm; Hanberger, Håkan

VERSION 1 – REVIEW

REVIEWER	Zhitao Yang Department of Emergency, Ruijin Hospital, Shanghai Jiao Tong University, School of medicine
REVIEW RETURNED	12-Jun-2021

GENERAL COMMENTS	1、 In Table 1 and Table, $P > 0.665$ and $P > 0.99$, similar to these, what dose it mean?2、 The data in this article are from 2015 to 2016. Why take this period? Why not use more recent data? I think recent data will bring more information to the reader.3、 Line 154-156, patients were deemed immunocompromised if prescribed immunosuppressive medication, it there a time limit for taking immunosuppressive medication?4、 Is the prehospital delay directly related to the delay of early hospital care? Is it caused by the patient's own cause?5、 Please recommend the author to analyze the use of antibacterial drugs and primary disease.
---

REVIEWER	Francesca Giovannenze Catholic University of the Sacred Heart
REVIEW RETURNED	23-Jun-2021

GENERAL COMMENTS	Summary The paper by Holmbom et al analyzed the impact of prehospital and early hospital care elements on 30-day mortality in community-onset BSI (CO-BSI). To this end, authors compared 195 non-survivors (cases) with 195 survivors (controls) with culture-confirmed CO-BSI in a matched case-control study, performing a logistic regression analysis to identify factors related to 30-day mortality. They found that a delay $>24h$ from first prehospital contact to hospital admission and inadequate empiric antibiotic therapy were independent risk factors for 30-day mortality in patients with CO-BSI, highlighting the importance of early identification and prompt management of sepsis in the community. This paper is In line with previous articles showing the association of delay in antibiotic administration with in-hospital mortality in patients with CO-BSI. Moreover, it is of great value in consideration of the
--

small available evidence on the impact of prehospital variables on patients' outcome. Research has mainly focused on early management of sepsis in the emergency department and in the hospital environment, while for CO-BSI the most delay, and consequently the most improvement opportunity, potentially lies in the prehospital setting. The manuscript is overall well written and the abundance of data regarding CO-BSI management in the pre-hospital phase is undoubtedly of great interest for the reader. Nevertheless, the article presents major flaws that should be addressed.

Major issues

- In lines 63-65 authors state that the increasing incidence of CO-BSI may be due to the increase in HCA-BSI of aged patients managed in the community setting. The distinction between CO-BSI and HCA-BSI is not clear. Please specify in the background and in the methods section that the definition of CO-BSI includes patients with both community-acquired (CA) BSI and community-onset healthcare-associated (CHA) BSI, since CHA-BSI have unique characteristics when compared to CA-BSI (Takeshita N, Kawamura I, Kurai H, Araoka H, Yoneyama A, Fujita T, Ainoda Y, Hase R, Hosokawa N, Shimanuki H, Sekiya N, Ohmagari N. Unique characteristics of community-onset healthcare-associated bloodstream infections: a multi-centre prospective surveillance study of bloodstream infections in Japan. J Hosp Infect. 2017 May;96(1):29-34. doi: 10.1016/j.jhin.2017.02.022. Epub 2017 Feb 27. PMID: 28377180.)

- In lines 72-81 authors describe the study setting, but the paragraph is erroneously placed in the background section. Please move it to the methods-study design and setting section.

- In lines 113-118 variables regarding the pre-hospital infection-related contacts collected in the CRF are listed, even though the data source is not clearly stated. Were data extracted from a regional database? Please specify the data source.

- The definition of appropriate antibiotic therapy is not clear and unique throughout the text. In lines 126-129 authors state that the correctness of empiric antibiotic treatment was judged by an expert group according to local recommendations, pathogen cultured and correction for renal function. In Table 1 "appropriate empirical antibiotics" and "appropriate antibiotics based on cultured bacteria" are presented as distinct variables. In Table 3 "inadequate empirical antibiotics" is a variable significantly associated to 30-day mortality at multivariable analysis: is this the "clinical" or "microbiological" inappropriateness? I understand that authors have tried to describe two distinct types of appropriateness with regard to empiric antibiotic therapy: a microbiological appropriateness (if the cultured pathogen was latter found to be susceptible to the antibiotic started empirically) and a global appropriateness, which in adjunction to microbiological susceptibility considers adherence to local recommendations and dosage issues. Please detail in the methods - definitions section the definition of "appropriate empiric antibiotic therapy" and of "microbiologically appropriate empiric antibiotic therapy". I suggest to use these two definitions throughout the text and tables.

- At lines 150-152, in the CO-BSI definition, authors state that patients who had received IV treatment or HD in the previous 30 days were excluded. Since this kind of infections are classified as community-onset Healthcare Associated BSI (CHA-BSI), this is in contrast with the definition of CO-BSI previously given. Authors

	should clearly define if patients with CHA-BSI are included in the study population. If so, even patients who receive IV treatment or HD should be included and the analysis repeated.  - According to reported results, the microbiological appropriateness of empiric antibiotic therapy does not significantly differ between survivors and non-survivors (lines 305-306) and subsequently its association with 30-day mortality is not investigated in the logistic regression analysis, while the rate of globally inappropriate antibiotic therapy is significantly higher in cases than in controls and is independently associated with 30-day mortality. In authors' opinion, what is the reason for this mismatch between global and microbiological appropriateness? Please provide an answer in the discussion section. - At lines 394-395 authors state that "The severity of illness on admission to hospital was greater in non-survivors and may be explained by prehospital delay". This conclusion is not coherent with the study results, since the correlation between prehospital delay and illness severity on admission has not been properly investigated. Please delete "and may be explained by prehospital delay". - The time of symptoms onset, which should be considered as the real time point for calculating hospital and therapy delay, is not available in this dataset. Add it as a major limitation in the Limitations paragraph. - Both a delay >24 hours in hospital admission (prehospital factor) and an inadequate empiric antibiotic therapy (early hospital factor) are found to be independent risk factors for 30-day all-cause mortality in the present study. Nevertheless, inadequate empiric antibiotic therapy is not mentioned at all in the conclusions. Please add it in the conclusions section. Minor issues  - Results in tables should be rounded to the same number of decimal places (i.e. 0,01 or 0,001). Please correct tables. - At lines 301-303 results on the median time to antibiotic therapy start are reported. Since when? It is included in the paragraph of "hospital data", so I understand that it refers to the time from hospital admission to antibiotic start. Please highlight that it refers to the median time from hospital admission to start of empiric antibiotic administration.
--	--

VERSION 1 – AUTHOR RESPONSE

Response to Reviewer 1

Dr Zhitao Yang, *Department of Emergency, Ruijin Hospital, Shanghai Jiao Tong University, School of medicine*

AU: Thank you for reading this paper, below are our responses to your questions and comments.

1. In Table 1 and Table, $P > 0.665$ and $P > 0.99$, similar to these, what dose it mean?

AU: Thank you. When SPSS rounds off the p-value to 1.000, even though p-value according to math can only be < 1.0 , it is written as > 0.99 . In Table 2, " $p > 0.665$ ", the $>$ sign has been incorrectly written. It now reads $p = 0.67$. This has been confirmed with our co-author MF who is a professional statistician

Table 2 now reads:

Line 261 (Table 2), $p = 0.67$

2. The data in this article are from 2015 to 2016. Why take this period? Why not use more recent data? I think recent data will bring more information to the reader.

AU: We agree. Data from 2019 and 2020 would have been more interesting. However, this is a retrospective study with matched cases and controls, involving extensive manual journal reviews, as shown in the attached CRF, and searches in several databases. With the resources at hand it has not been possible to conduct such a study with more recent data.

3. Line 154-156, patients were deemed immunocompromised if prescribed immunosuppressive medication, is there a time limit for taking immunosuppressive medication?

AU: We agree, the definition should be clarified. We have added "when admitted to the hospital". The text now reads:

Lines 160-162: "Immunosuppression: Patients were deemed immunocompromised if prescribed immunosuppressive medication or had a code number (D80-D89) for an immunosuppressive disease according to the International Classification of Diseases, 10th edition (ICD-10) when admitted to the hospital."

4. Is the prehospital delay directly related to the delay of early hospital care? Is it caused by the patient's own cause?

AU: Unfortunately, there was no available information on patient delay because time at onset of symptoms was not systematically registered in the patient records. This is mentioned in the paragraph on strengths and limitations of the study, Lines 48-49.

Lines 48-49: "Data regarding patient delay *i.e.*, time from onset of symptoms to contact with prehospital or hospital care, were unavailable."

According to our results, we could not show that prehospital delay was related to delay in early hospital care. No significant difference was seen between non-survivors and survivors regarding paramedic alarm times or time before antibiotics given in hospital (and presumably other treatments also). Table 1 shows time to antibiotics given in hospital, where no difference can be seen between non-survivors and survivors. Furthermore, no difference was seen in early hospital care according to the variables available in this retrospective study (we mention this in lines 295-299 and 343-345).

Lines 295-299: "Antibiotic treatment was initiated in 96% of non-survivors and 99.5% of survivors, ($p=0.04$). The median time from hospital admission to start of empirical antibiotic did not differ between the groups, non-survivors 2.8 hours (1.4-5.3) and survivors 3.0 hours (1.4-6.2). Antibiotic administration was started within one hour in 18% of non-survivors and 16% of survivors ($p=0.48$)."

Lines 343-345: "Although initial care at the hospital and time to antibiotic treatment were comparable between non-survivors and survivors (for some variables even better among non-survivors) this was not enough to compensate for the negative effect of prehospital delay."

On the other hand, there was a difference between non-survivors and survivors regarding the delay between first contact due to infection with a healthcare facility and arrival at the hospital *i.e.*, before early hospital care was given.

To make things clearer, we have added a sixth table to the supplementary file showing times for ambulance transport and time to empirical antibiotic treatment based on triage score (RETTS). We have also added a comment in the results, lines 295-299 and in the discussion, lines 451-454 and 439-445. The text now reads:

Table 6 supplementary file: Early hospital care - Time for ambulance transport and time to empirical antibiotic treatment based on triage score (RETTS)

We have added nor in the case of sub-classification according to triage score (RETTS)
Lines 295-299:

Antibiotic treatment was initiated in 96% of non-survivors and 99.5% of survivors, ($p=0.04$). The median time from hospital admission to start of empirical antibiotic did not differ between the groups, non-survivors 2.8 hours (1.4-5.3) and survivors 3.0 hours (1.4-6.2), nor in the case of sub-classification according to triage score (RETTS). Antibiotic administration was started within one hour in 18% of non-survivors and 16% of survivors ($p=0.48$).

Lines 451-454: Data regarding patient delay *i.e.*, time from onset of symptoms to contact with prehospital or hospital care should be included when calculating hospital and therapy delay. However, these were unavailable since patient delay was not systematically registered in the patient records.

Lines 439-445: The only time parameters for early hospital care in this study were time to empirical antibiotic treatment and time for ambulance transport. Presumably, time to antibiotic treatment also implies time to early hospital treatment. Furthermore, time to early hospital care is usually based on severity of illness (RETTS and NEWS 2) when admitted to hospital. Time to antibiotic treatment based on RETTS did not differ between the groups. This study was not designed to validate the time to care based on triage level, RETTS or NEWS 2 correctness.

5. Please recommend the author to analyze the use of antibacterial drugs and primary disease.

AU: In the results section, line 279-282, we describe the suspected primary focus of the infection.

Line 279-282: "The most common infection focus was a urinary tract infection (24%) followed by infection with unknown focus (23%), and respiratory tract infection (22%). There were no differences between non-survivors and survivors regarding primary focus of infection (Supplementary File, Table 3)."

Regarding use of antibacterial drugs, we have added a seventh table to the supplementary file showing listed empirical antibiotics. A comment has also been added in the results section, line 302-303.

The text now reads:

Table 7 Supplementary File: Empirical antibiotics used.

Line 302-303: The empirical antibiotics used are listed in Supplementary File, Table 7,

Response to Reviewer 2

Dr. Francesca Giovannenze, Catholic University of the Sacred Heart

"The paper by Holmbom et al analyzed the impact of prehospital and early hospital care elements on 30-day mortality in community-onset BSI (CO-BSI). To this end, authors compared 195 non-survivors (cases) with 195 survivors (controls) with culture-confirmed CO-BSI in a matched case-control study, performing a logistic regression analysis to identify factors related to 30-day mortality. They found that a delay >24h from first prehospital contact to hospital admission and inadequate empirical antibiotic therapy were independent risk factors for 30-day mortality in

patients with CO-BSI, highlighting the importance of early identification and prompt management of sepsis in the community. This paper is in line with previous articles showing the association of delay in antibiotic administration with in-hospital mortality in patients with CO-BSI. Moreover, it is of great value in consideration of the small available evidence on the impact of prehospital variables on patients' outcome. Research has mainly focused on early management of sepsis in the emergency department and in the hospital environment, while for CO-BSI the most delay, and consequently the most improvement opportunity, potentially lies in the prehospital setting. The manuscript is overall well written and the abundance of data regarding CO-BSI management in the pre-hospital phase is undoubtedly of great interest for the reader. Nevertheless, the article presents major flaws that should be addressed”.

AU: Thank you for reading this paper, below are our responses to your questions and comments.

1. In lines 63-65 authors state that the increasing incidence of CO-BSI may be due to the increase in HCA-BSI of aged patients managed in the community setting. The distinction between CO-BSI and HCA-BSI is not clear. Please specify in the background and in the methods section that the definition of CO-BSI includes patients with both community-acquired (CA) BSI and community-onset healthcare-associated (CHA) BSI, since CHA-BSI have unique characteristics when compared to CA-BSI (Takeshita N, Kawamura I, Kurai H, Araoka H, Yoneyama A, Fujita T, Ainoda Y, Hase R, Hosokawa N, Shimanuki H, Sekiya N, Ohmagari N. Unique characteristics of community-onset healthcare-associated bloodstream infections: a multi-centre prospective surveillance study of bloodstream infections in Japan. *J Hosp Infect.* 2017 May;96(1):29-34. doi: 10.1016/j.jhin.2017.02.022. Epub 2017 Feb 27. PMID: 28377180.)

AU: We agree, it is important not to use hospital acquired (HA), community-acquired (CA), community-onset (CO), community-onset healthcare-associated (CHA) and healthcare-associated (HCA) synonymously, nor bacteraemia and sepsis, because microbial aetiology, empirical antibiotic treatment, and outcome all differ [1-3]. An important challenge is the dramatic shift in healthcare delivery in recent years, with complex medical treatments such as haemodialysis and parenteral antibiotics now being given in the community setting. As a result, a community-onset infection may, by definition, be a hospital-associated/acquired infection. Community-onset infection is usually defined as “one occurring in the outpatient setting or first identified (cultures drawn) within 2 days (<48h) after hospital admission”. Hospital-acquired is defined as “an infection where a positive culture is first identified 2 or more days (≥48h) after hospital admission or within 2 days (<48h) after hospital discharge” [4]. Specialised care in the community setting has increased, and the definition of community-onset BSI has been further subclassified into healthcare-associated (HCA/CHA) *i.e.*, patients with ongoing or significant prior healthcare contact, all others are classified as community-acquired (CA) [2, 5]. In Sweden, healthcare (primary and specialised) includes community care, private medicine and self-medication *i.e.*, by the patient or a relative where data are unavailable. This makes it impossible, in a retrospective study, to classify community-associated infection, be it CA-BSI or HCA/CHA-BSI according to the criteria of Friedman *et al* where to be able to classify an infection as HCA/CHA-BSI requires at least one of the following: a) patient recently hospitalised; b) patient recently received specialised medical care at home; c) patient recently attended a hospital-based clinic or haemodialysis unit; and d) patient is a nursing home resident. Patients not having any these criteria are deemed community-acquired BSI.

The aim was to study CA-BSI but we cannot with certainty classify the cohort as CA-BSI according to the criteria above because of care received in the community where data were unavailable as mentioned above. Therefore, we chose to use a wider concept, CO-BSI. However, after consideration by the author group, we decided it is more accurate to use the term CA-BSI, but at the same time clarify in the limitation section that it is possible that HCA/CHA-BSI may have occurred in the study population. According to our definition, hospital- and healthcare-related infections were excluded as far as possible. After a reviewed of our data, no readmissions were included according to the definition below. We have revised and replaced CO-BSI with CA-BSI.

The definition is rewritten and now reads:

154-158: Community-acquired bloodstream infection (CA-BSI): A positive blood culture obtained within 48 hours after hospital admission. Cultures from readmission within 30 days were excluded. Furthermore, patients who had received intravenous treatment or advanced nursing care at home such as parenteral nutrition, haemodialysis, or intravenous chemotherapy, within the previous 30 days were excluded.

The title is rewritten and now reads:

Line 1-2: *“Prehospital delay is an important risk factor for mortality in community-acquired bloodstream infection (CA-BSI): A matched case-control study”*

In the limitation section, the text now reads:

Line 446-450: Community healthcare in Sweden has several healthcare providers i.e., care provided by the community, private medicine and/or self-medication (by patient or relative). This result in unavailable data and it is possible that some community-acquired healthcare-associated BSIs may have occurred in the study population, even though hospital and healthcare-related infections were excluded as far as possible.

- 1) De Bus, L., et al., *Microbial etiology and antimicrobial resistance in healthcare-associated versus community-acquired and hospital-acquired bloodstream infection in a tertiary care hospital. Diagn Microbiol Infect Dis*, 2013. **77**(4): p. 341-5.
- 2) Friedman, N.D., et al., *Health care--associated bloodstream infections in adults: a reason to change the accepted definition of community-acquired infections. Ann Intern Med*, 2002. **137**(10): p. 791-7.
- 3) Lenz, R., et al., *The distinct category of healthcare associated bloodstream infections. BMC Infect Dis*, 2012. **12**: p. 85.
- 4) Morin, C.A. and J.L. Hadler, *Population-based incidence and characteristics of community-onset Staphylococcus aureus infections with bacteremia in 4 metropolitan Connecticut areas, 1998. J Infect Dis*, 2001. **184**(8): p. 1029-34.
- 5) Diekema, D.J., et al., *Epidemiology and outcome of nosocomial and community-onset bloodstream infection. J Clin Microbiol*, 2003. **41**(8): p. 3655-60.

2. In lines 72-81 authors describe the study setting, but the paragraph is erroneously placed in the background section. Please move it to the methods-study design and setting section.

AU: This has been carried out. Lines 96-104

3. In lines 113-118 variables regarding the pre-hospital infection-related contacts collected in the CRF are listed, even though the data source is not clearly stated. Were data extracted from a regional database? Please specify the data source.

AU: The main data source was based on culture-confirmed BSI in adults 2015-2016. This is described in the methods section. Data from infection-related contacts were obtained by reviewing the medical records of non-survivors and survivors.

The text now reads:

Lines 121-123: For non-survivors and survivors data on all infection-related contacts with a healthcare provider (telephone contact or physical contact) ≤ 7 days prior to a positive blood culture were collected by reviewing medical records.

4. The definition of appropriate antibiotic therapy is not clear and unique throughout the text. In lines 126-129 authors state that the correctness of empirical antibiotic treatment was judged by an expert group according to local recommendations, pathogen cultured and correction for renal function. In Table 1 “appropriate empirical antibiotics” and “appropriate antibiotics based on cultured bacteria” are presented as distinct variables. In Table 3 “inadequate empirical antibiotics” is a variable significantly associated to 30-day mortality at multivariable analysis: is this the “clinical” or “microbiological” inappropriateness? I understand that authors have tried to describe two distinct types of appropriateness with regard to empirical antibiotic therapy: a microbiological appropriateness (if the cultured pathogen was latter found to be susceptible to the antibiotic started empirically) and a global appropriateness, which in adjunction to microbiological susceptibility considers adherence to local recommendations and dosage issues. Please detail in the methods - definitions section the definition of “appropriate empirical antibiotic therapy” and of “microbiologically appropriate empirical antibiotic therapy”. I suggest to use these two definitions throughout the text and tables.

AU: This has been carried out. We have added the definition of “appropriate empirical antibiotic therapy” and “microbiologically appropriate empirical antibiotic therapy” and we use these two definitions throughout the text as recommended. Regarding “inadequate empirical antibiotics”, it has been renamed to inappropriate empirical antibiotic therapy and we have added a footnote to Table 3.

The text in Methods section and Table 3 now reads:

Footnote Table 3 (line 330-331): *** Clinical inappropriateness defined by local recommendations, suspected source of infection, severity of illness, dosage, and correction for renal function.

Line 176-177: Appropriate empirical antibiotic therapy: Defined by local recommendations, suspected source of infection, severity of illness, dosage, and correction for renal function [34,35].

Line 178-179: Microbiologically appropriate empirical antibiotic therapy: Defined according to cultured pathogen and its susceptibility patterns [34-36]

5. At lines 150-152, in the CO-BSI definition, authors state that patients who had received IV treatment or HD in the previous 30 days were excluded. Since this kind of infections are classified as community-onset Healthcare Associated BSI (CHA-BSI), this is in contrast with the definition of CO-BSI previously given. Authors should clearly define if patients with CHA-BSI are included in the study population. If so, even patients who receive IV treatment or HD should be included and the analysis repeated.

AU: Please see answer to question 1 for clarification of definitions

6. According to reported results, the microbiological appropriateness of empirical antibiotic therapy does not significantly differ between survivors and non-survivors (lines 305-306) and subsequently its association with 30-day mortality is not investigated in the logistic regression analysis, while the rate of globally inappropriate antibiotic therapy is significantly higher in cases than in controls and is independently associated with 30-day mortality. In authors’

opinion, what is the reason for this mismatch between global and microbiological appropriateness? Please provide an answer in the discussion section.

AU: We agree, this mismatch is not easy to explain. It is probable that correct empirical antibiotic is not the only important factor in initial management of BSI. This is discussed in lines 406-411.

Lines 406-411: "Appropriate empirical antibiotic treatment according to local guidelines was more common among survivors ($p < 0.01$), while appropriate antibiotic according to subsequent blood cultures was not. This illustrates that not only the administration of correct empirical antibiotics according to recommendations but also a holistic view of the patient, including suspected focus and need of source control, are both important parts of initial management. However, this study was not designed to evaluate the correctness of antimicrobial therapy."

The text is rewritten:

Line 406-411: Appropriate empirical antibiotic therapy according to local guidelines was more common among survivors ($p < 0.01$), while microbiologically appropriate empirical antibiotic therapy was not. This could indicate that not only correct empirical antibiotics is important, but that also a holistic view of the patient, including suspected focus of infection and need of source control, are important parts of early management.

7. At lines 394-395 authors state that "The severity of illness on admission to hospital was greater in non-survivors and may be explained by prehospital delay". This conclusion is not coherent with the study results, since the correlation between prehospital delay and illness severity on admission has not been properly investigated. Please delete "and may be explained by prehospital delay".

AU: We use the word "may" to highlight the uncertainty of the statement. Prehospital delay (>24h) and in-hospital SOFA-score were related to mortality in the multivariable analysis. In the univariate data, severity of illness on admission was greater in non-survivors and was also associated with mortality in the multivariable analysis as mentioned above. The assumption that prehospital delay may have increased the severity of illness thus seems reasonable. We have rewritten the text to clarify this.

The text now reads:

Line 392-394: The severity of illness on admission to hospital was greater in non-survivors and there are several factors that might explain this. One factor could be a longer prehospital delay as observed in this study.

8. The time of symptoms onset, which should be considered as the real time point for calculating hospital and therapy delay, is not available in this dataset. Add it as a major limitation in the Limitations paragraph.

AU: This has been carried out. The text now reads:

Line 451-454: Data regarding patient delay *i.e.*, time from onset of symptoms to contact with prehospital or hospital care, should be included when calculating hospital and therapy delay. However, these were unavailable since patient delay was not systematically registered in the patient records.

9. Both a delay >24 hours in hospital admission (prehospital factor) and an inadequate empirical antibiotic therapy (early hospital factor) are found to be independent risk factors for 30-day all-cause mortality in the present study. Nevertheless, inadequate empirical antibiotic therapy is not mentioned at all in the conclusions. Please add it in the conclusions section.

AU: Thank you, this has been carried out. The text now reads:

Line 30-31: Conclusion: Prehospital delay and inappropriate empirical antibiotic therapy were found to be important risk factors for 30-day all-cause mortality associated with CA-BSI.

Line 457-461: In this case-control study, prehospital delay and inappropriate empirical antibiotic therapy were found to be important risk factor for 30-day all-cause mortality associated with CA-BSI. Effective guidelines to aid recognition of patients likely to develop sepsis in prehospital and early hospital care, and an effective prehospital medical advice service, could help to make significant progress in the care of patients with bloodstream infection.

- 10.** Results in tables should be rounded to the same number of decimal places (i.e., 0,01 or 0,001). Please correct tables.

AU: The numbers in the tables are now rounded to two decimal places. The tables now read: (see **Line 223** (Table 1), **Line 261** (Table 2), **Line 329** (Table 3) and **Supplementary Tables**

- 11.** At lines 301-303 results on the median time to antibiotic therapy start are reported. Since when? It is included in the paragraph of “hospital data”, so I understand that it refers to the time from hospital admission to antibiotic start. Please highlight that it refers to the median time from hospital admission to start of empirical antibiotic administration.

AU: This has been carried out. The text now reads:

Lines 295-297: Antibiotic treatment was initiated in 96% of non-survivors and 99.5% of survivors, ($p=0.04$). The median time from hospital admission to start of empirical antibiotic did not differ between the groups, non-survivors 2.8 hours (1.4-5.3) and survivors 3.0 hours (1.4-6.2).

VERSION 2 – REVIEW

REVIEWER	Zhitao Yang Department of Emergency, Ruijin Hospital, Shanghai Jiao Tong University, School of medicine
REVIEW RETURNED	28-Sep-2021

GENERAL COMMENTS	The data in this article are from 2015 to 2016. Suggest use more recent data?
---

REVIEWER	Francesca Giovannenze Catholic University of the Sacred Heart
REVIEW RETURNED	18-Sep-2021

GENERAL COMMENTS	All changes requested were made and all points addressed. In particular you clarified that community-onset healthcare-associated BSI were excluded from the study population and you better explained the difference between appropriate empirical antibiotic therapy and microbiological appropriate empirical antibiotic therapy.
---

VERSION 2 – AUTHOR RESPONSE

Reviewer: 2

Dr. Francesca Giovannenze, Catholic University of the Sacred Heart Comments to the Author:

1. All changes requested were made and all points addressed. In particular you clarified that community-onset healthcare-associated BSI were excluded from the study population and

you better explained the difference between appropriate empirical antibiotic therapy and microbiological appropriate empirical antibiotic therapy.

AU: Thank you for your positive comments to our revised manuscript. Your thoughtful and constructive comments in reviewing process have improved and strengthened our manuscript.

Reviewer: 1

Dr. Zhitao Yang , Department of Emergency, Ruijin Hospital, Shanghai Jiao Tong University, School of medicine Comments to the Author:

1. The data in this article are from 2015 to 2016. Suggest use more recent data?

AU: Thank you very much for reviewing our manuscript and for the comments. We agree with you that more recent data would have been of great interest. However, this is a retrospective study with matched case and controls. The study was launched 2017 and in June 2018 the bureaucratic process regarding access to the dataset was ended. Data collection started in august 2018, involved extensive manual journal reviews (as shown in the attached CRF) and this work was an extremely time-consuming process. The analysis phase started in May 2019 and completed at the end of 2019. The ongoing COVID-19 pandemic has affected and curtailed clinical research, or redirected resources/research to COVID-19. Most clinical research have been paused in our area and we (as physicians') have been focusing on improving the patient-care. At the end of 2020, the research process starting again and we submitted the first draft in April 2021 (the manuscript has not been previously submitted). With the resources at hand and an ongoing pandemic, it has not been possible to complete the study earlier.